# Olfactory Training in Post-COVID-19 Persistent Olfactory Disorders: Value Normalization for Threshold but Not Identification

**DOI:** 10.3390/jcm11123275

**Published:** 2022-06-08

**Authors:** Clair Vandersteen, Magali Payne, Louise-Émilie Dumas, Élisa Cancian, Alexandra Plonka, Grégoire D’Andréa, David Chirio, Élisa Demonchy, Karine Risso, Florence Askenazy-Gittard, Charles Savoldelli, Nicolas Guevara, Philippe Robert, Laurent Castillo, Valeria Manera, Auriane Gros

**Affiliations:** 1Institut Universitaire de la Face et du Cou, Centre Hospitalier Universitaire, Université Côte d’Azur, 31 Avenue de Valombrose, 06100 Nice, France; cancian.e@chu-nice.fr (É.C.); dandreagregoire3@gmail.com (G.D.); savoldelli.c@chu-nice.fr (C.S.); guevara.n@chu-nice.fr (N.G.); castillo.l@chu-nice.fr (L.C.); 2Laboratoire CoBTeK, Université Côte d’Azur, 06100 Nice, France; magali-payne06@orange.fr (M.P.); dumas.le@pediatrie-chulenval-nice.fr (L.-É.D.); alexandra.pl@gmail.com (A.P.); askenazy.f@pediatrie-chulenval-nice.fr (F.A.-G.); philippe.robert@univ-cotedazur.fr (P.R.); valeria.manera@unice.fr (V.M.); auriane.gros@univ-cotedazur.fr (A.G.); 3Département d’Orthophonie de Nice (DON), UFR Médecine, Université Côte d’Azur, 06107 Nice, France; 4Hôpitaux Pédiatriques de Nice CHU-LENVAL, Centre Hospitalier Universitaire, Université Côte d’Azur, 57 Avenue de la Californie, 06200 Nice, France; 5Institut NeuroMod, INRIA Centre de Recherche Sophia Antipolis, Université Côte d’Azur, 2004 Route des Lucioles, 06902 Valbonne, France; 6Service Clinique Gériatrique du Cerveau et du Mouvement, Centre Hospitalier Universitaire de Nice, 06000 Nice, France; 7Département de Médecine Infectiologique, Hôpital de l’archet, Centre Hospitalier Universitaire, Université Côte d’Azur, 151 Route de Saint-Antoine, 06200 Nice, France; chirio.d@chu-nice.fr (D.C.); demonchy.e@chu-nice.fr (É.D.); risso.k@chu-nice.fr (K.R.)

**Keywords:** COVID-19, olfaction disorders, olfactory training, parosmia

## Abstract

(1) Background: Persistent post-viral olfactory disorders (PPVOD) are estimated at 30% of patients one year after COVID-19 infection. No treatment is, to date, significantly effective on PPVOD with the exception of olfactory training (OT). The main objective of this work was to evaluate OT efficiency on post-COVID-19 PPVOD. (2) Methods: Consecutive patients consulting to the ENT department with post-COVID-19 PPVOD were included after completing clinical examination, the complete Sniffin’ Stick Test (TDI), the short version of the Questionnaire of olfactory disorders and the SF-36. Patients were trained to practice a self-olfactory training with a dedicated olfactory training kit twice a day for 6 months before returning to undergo the same assessments. (3) Results: Forty-three patients were included and performed 3.5 months of OT in average. We observed a significant TDI score improvement, increasing from 24.7 (±8.9) before the OT to 30.9 (±9.8) (*p* < 0.001). Based on normative data, a significant increase in the number of normosmic participants was observed only for the threshold values (*p* < 0.001). Specific and general olfaction-related quality of life improved after the OT. (4) Conclusions: Olfactory function appeared to improve only in peripheral aspects of post-COVID-19 PPVOD after OT. Future controlled studies must be performed to confirm the OT role and justify new therapeutic strategies that may focus on the central aspects of post-COVID-19 PPVOD.

## 1. Introduction

Long persistent olfactory complaint is widely reported after an acute, mild or moderate, COVID-19 infection. Indeed, a complete but subjective olfaction recovery is only reported in 40% to 63% [1,2,3] and 70% [4] of patients, respectively, 6 and 12 months after COVID-19. Interestingly, olfactory psychophysical tests results were better than subjective smell assessments, showing 73% to 95% normosmic patients after 6 months [5,6]. Some authors suggested that these remaining olfactory complaining patients, with no recovery 18 months after acute phase [7], could be permanently impaired. Parosmia is the main qualitative dysosmia associated with COVID-19 olfactory recovery and occurs in 18% to 49% [3,7,8] of patients 2.5 months after the acute phase of infection. Parosmia affects 20% of normosmic patients [3] and contributes to the discrepancy between subjective impairment and olfactory psychophysical tests.

Long-lasting olfactory loss leads to a quality of life(QoL) [8] impairment, bad diet habits [9], changes in social and personal relations [10], psychiatric disorders (such as depression [11]), anxiety or anorexia [12] and its nutritional consequences [13], cognitive impairment [10] or increase in the hazardous event incidence [14]. Thus, they must be managed. Many treatments [15] have been tested to obtain an olfaction recovery without significant results, including vitamins, minerals and corticosteroids, as well in COVID-19 as reported in a recent Cochrane living review [16]. 

Olfactory training (OT), as described by Hummel et al. [17,18], remains the best treatment for persistent post-viral olfactory disorder (PPVOD) [19]. Indeed, in PPVOD, OT systematically improved the Minimal Clinically Important Difference (MCID) of the complete Sniffin’ Stick Test score (TDI; +6 [17]) in 30% to 68% of cases (OR = 2.77) [19]. However, the effectiveness of OT on post-COVID-19 PPVOD patients remains unknown. The primary objective of this work was to evaluate the olfaction recovery of patients who performed OT in a post-COVID-19 PPVOD.

## 2. Materials and Methods

The study was approved by the institutional review board of the Nice University Hospital (CNIL number: 412) and registered with a ClinicalTrials.gov number (ID: NCT04799977). Since March 2020, we prospectively enrolled patients in the ENT division of Nice University Hospital until December 2020. All were contaminated by COVID-19 with persistent olfactory disorders lasting more than 6 weeks (3 to 15 months). Patients where mainly self-referred or referred by general practitioners or colleagues. Patients had either a RT-PCR-proven SARS-CoV-2 diagnosis or a CT-proven SARS-CoV-2 diagnosis secondarily confirmed by serology. 

We retrospectively extracted the patients’ demographic data and clinical features, including subjective taste impairment, subjective olfactory impairment (qualitative and quantitative dysosmia), the visual analogue scale (VAS) for the subjective assessment of olfactory recovery (ranging from 0% to 100%), weight (measured at home in the previous week on a personal scale), nasofibroscopy (assessing nasal cavity patency and differential diagnosis), evaluation of olfactory loss using Sniffin’ Sticks Test^®^ (SST; Medisense, Groningen, The Netherlands) [20], completion of the French short version of Questionnaire of Olfactory Disorders (Short-QOD-NS) [21] and completion of the French SF-36 [22]. 

Patients were initially trained in the daily use of an OT protocol (detailed below) for 6 months. A second consultation (scheduled at 6 months ± 15 days) after OT allowed a second assessment of all the same elements except for nasofibroscopy. We calculated OT compliance through a score based on the following formula: (((number of OT weeks performed/24)—(number of OT sessions missed per week/14))/number of OT weeks performed) × 100. 

The “Number of OT weeks performed/24” represents the week proportion of OT performed compared to the protocol instruction explained to patients. The “Number of OT sessions missed per week/14” represents the week proportion of OT sessions missed per week as a week include 14 sessions (2 per day). The formula part “(number of OT weeks performed/24)—(number of OT sessions missed per week/14)” represent the real performed olfactory training proportion compared to total olfactory training weeks (“number of OT weeks performed”). The result of the formula reflects compliance of patient to the olfactory training protocol.

### 2.1. Objective Olfactory Dysfunction

Olfactory function was assessed using the Sniffin’ Sticks test, a validated psychophysical test that includes a phenyl-ethyl alcohol (PEA) odor Threshold detection (T), an odor Discrimination (D) and an odor Identification (I) test. The detailed procedure was previously detailed [23]. The results from the three tests were summed up to a composite score, the “TDI”. As described by the last update of TDI normative values [20], normosmia, hyposmia and anosmia were, respectively, defined by TDI ≥ 30.75, 30.5 ≥ TDI ≥ 16.25 and TDI ≤ 16. Concerning isolated T, D and I values, normal and reduced olfactory function, related to gender and age, were, respectively, defined as ≥10th percentile and <10th percentile subdimension scores based on [20].

### 2.2. Olfactory Training

OT was based on the Hummel et al. protocol [17], whose regenerative properties on olfactory neurons, olfactory cortex connectivity and olfactory scores are widely reported in the literature [15,19,24]. The protocol was explained to the patients. This involved a 6-month olfactory training with daily odor exposure, twice a day (two sessions) with two different random odors from the kit in the morning and the evening (four different odors per day). We decided to run the olfactory training for 6 months as it was described [25] to be more effective than the 12 week previously described protocol [17], especially since odors are renewed every 3 months [15]. 

Patients were instructed, once all the odors had been used at least once, to attempt to recognize them by blindly sniffing them. To improve compliance and the ludic aspects of the OT, we used other odors than the four common ones (Phenylethyl alcohol [Rose], Eucalyptol [Eucalyptus], Citronellal [Citronella] and Eugenol [Clove]) [17]. We used an olfaction training kit that was produced specifically for this purpose by a local industry, including 11 small pots of scented wax (10 g), impregnated with 15% of dill, thyme, cinnamon, cloves, coriander leaf, vinegar, cumin, lavender, coffee, vanilla or mint. We used different types of odors because no significant difference in olfaction improvement was reported using simple or complex odors or combinations of both [26].

### 2.3. Olfactory Quality of Life

Olfactory QoL was assessed using the French validated Short-QOD-NS [21] self-questionnaire (2 min), which is based on negative statements from the Questionnaire of Olfactory Disorders (QOD) but is shorter, allowing an increase in the response rate and reducing the patient’s mental load when completing the questionnaire. These negative statements of QOD were shown to be more correlated with the results of psychophysical olfactory tests (SST) [27]. The Short-QOD-NS [28] includes the seven most relevant questions related to social aspects (*n* = 3), eating (*n* = 2), anxiety (*n* = 1) and annoyance (*n* = 1) following an olfactory loss. The score ranges from 0 to 21 (21 meaning there is no impairment).

The 36-item form health survey (SF-36) is one of the most widely used generic questionnaires, validated in French by Perneger et al. [29] and used here to evaluate general QoL. The SF-36 questionnaire consists of 36 self-administered questions (5 min) divided into eight domains, covering both physical and mental health. The physical component summary covers four subdomains as initially described: physical functioning, role physical functioning, bodily pain and general health. The mental component summary covers four other domains: vitality, emotional functioning, social functioning and mental health. The sum of the score is calculated for each domain and scaled to 100. High scores indicate good QoL, while low scores indicate low QoL.

### 2.4. Statistical Analysis

To explore the evolution of quantitative variables (e.g., TDI scores) before and after the OT, we employed Wilcoxon signed-rank tests, as most of the data did not follow a normal distribution (as confirmed by Shapiro–Wilks tests). To compare the evolution of binary variables (e.g., the presence of parosmia and phantosmia), we employed the McNemar test. To compare quantitative variables (such as TDI scores) between different groups of participants (e.g., participants with vs. without parosmia) we employed Mann–Whitney tests. Non-parametric correlations (Spearman rho) were employed to investigate correlations between treatment compliance and improvement in TDI scores. All results were considered statistically significant for a bilateral alpha level of 0.05.

## 3. Results

### 3.1. Demographic and Clinical Features

Forty-three patients were included in the study. The demographic and clinical initial features are reported in Table 1. None of the enrolled patients reported a history of previous olfactory impairment.

Patients were seen 5.8 ± 3.2 months and 11 ± 3.7 months after COVID-19 infection, respectively, at the first and the second (after OT) visit. Twenty-eight patients received a COVID-19 related treatment (of which 6 and 2 took, respectively, oral and nasal corticosteroids from 1 to 3 weeks). Among people who had a medical history of self-immune diseases, two had Crohn’s disease, and one had ankylosing spondylitis. Some had a medical history of neurological diseases, two had epilepsy under specific medications, and one had a stroke during childhood with no sequelae. Nasofibroscopies found no obstructive pathologies in the olfactory cleft.

On average, patients lost weight between the two visits before and after OT, going from 69.8 ± 13.3 kg to 66.7 ± 20.1 kg; however, the weight reduction was not statistically significant (Z = −0.88, *p* = 0.378). VAS Subjective olfactory recovery significantly increased from 34.6 ± 26.5% to 57.9 ± 31.1% (Z = −4.71, *p* < 0.001). A slight, but not significant, decrease from 37 to 32 patients (74.4%) of chemosensorial complaints was reported after OT (*p* = 0.125), with 30 (69.8%) and 5 (11.6%) patients who still suffered from flavors and/or taste loss after the OT.

### 3.2. Olfactory Training Results

#### 3.2.1. Compliance

Even if seen after 6 months, patients only performed 14.5 ± 9 weeks (~3.5 months) of OT on average, missing 0.8 ± 1 entire day per week and 3 ± 3 sessions per week. Patients most often reported forgetting to do the training, slowing down the use of the kit due to despondency or not finding the time to do the (short) rehabilitation sessions. The average compliance ratio was 53 ± 33%, ranging from 4% to 110% (for people that comleted more than 24 weeks). There was no significant correlation between OT compliance and TDI evolution or subjective olfaction evaluation recovery—VAS or improvements in quality of life (all ps > 0.101).

#### 3.2.2. TDI

There was a significant improvement in the mean TDI score (Z = −4.71, *p* < 0.001), which increased from 24.7 (±8.9) before the OT to 30.9 (±9.8) after the OT. A significant change in the number of participants categorized as anosmic, hyposmic and normosmic before and after the training was found (Chi2 = 25.7, *p* < 0.001). Specifically, the number of anosmic participants decreased from 10 (23.3%) to 5 (11.6%); the number of hyposmic decreased from 22 (51.2%) to 11 (25.6%); and the number of normosmic participants increased from 11 (25.6%) to 27 (62.8%). These results are graphically reported in Figure 1. 

As compliance to the OT protocol was not ideal for several participants, and most OT-validated protocols suggest a minimum of 8 weeks training duration [15], we compared the TDI scores of patients who completed less than 2 months (*n* = 18) of OT to those who completed more than 2 months (*n* = 25). We observed a better, but not significant (*p* = 0.089), TDI score improvement in patients who completed more than 2 months of OT, respectively, 4 ± 5 (*n* = 18) vs. 7.6 ± 7.5 (*n* = 25). Moreover, T (4.2 ± 4.4) was the most improved subdimension compared to I (1.9 ± 2.6) and D (1.5 ± 3.2).

The T and I scores significantly improved after the OT (from 4.9 ± 3.9 to 8.7 ± 5.2, Z = −4.67, *p* < 0.001; and from 9.4 ± 4.1 to 11.0 ± 3.4, Z = −3.60, *p* < 0.001, respectively). No significant evolution of the D score was observed (from 10.4 ± 3.0 to 11.2 ± 3.3, Z = −1.60, *p* = 0.110). The improvement in T was significantly large than the improvement in D (Z = −4.1, *p* < 0.001) and I (Z = −2.7, *p* = 0.007). T Improvement was significantly correlated with subjective recovery evaluation (VAS, *p* = 0.039). 

Concerning the evolution of the number of participants that reached the norms for T, D and I (based on normative data divided by sex and age), a significant increase in normalized value was found only for the T (McNemar test, *p* < 0.001) and not for the D (McNemar test, *p* = 0.774) or the I (McNemar test, *p* = 0.388). Based on age- and sex-normalized values, the number of participants who had normal T, D and I before and after the OT are presented in Figure 2. No clinical, medical history, treatment or compliance predictive value was significantly correlated to better psychophysical tests results.

#### 3.2.3. Qualitative Dysosmia

The number of participants reporting the presence of parosmia increased significantly from 8 (18.6%) to 27 (62.8%) after the OT (McNemar test, *p* < 0.001), with only one participant (2%) that fully recovered after the OT and 20 participants (46.5%) that developed parosmia after the OT. At the end of the OT, participants presenting parosmia showed lower identification scores (U = 122, *p* = 0.018). Patients with parosmia were significantly less likely to lose weight (Z = −2.4; *p* = 0.013). No significant difference in the number of participants reporting phantosmia was found (nine participants (20.9%) before the OT and 12 (27.9%) after the OT, *p* = 0.581; eight subjects (18.6%) developed phantosmia after the OT, and 5 (11.6%) recovered after the OT). No predictive factor was significantly associated with qualitative dysosmia evolution.

### 3.3. Quality of Life

The results of the SF36 and the Short-QOD-NS questionnaires are reported in Table 2. Concerning the SF36, significant improvements after the OT were obtained in the subdomains assessing physical functioning (*p* = 0.009), social functioning (*p* = 0.013, emotional role (*p* = 0.049), vitality (*p* = 0.023) and general health perception (*p* = 0.045). For the Short-QOD-NS, significant improvements after the OT were observed for the total score (*p* < 0.001) and all the subdomains, namely the social (*p* = 0.001), the food (*p* = 0.036), the anxiety (*p* = 0.020) and the annoyance (*p* = 0.020) subdomains. Short-QOD-NS improvement was significantly correlated to a TDI improvement after OT (*p* = 0.008).

## 4. Discussion

Persistent post-COVID olfactory loss is becoming a social issue as millions of people worldwide are affected. OT is, for the moment, the only therapeutic hope for post-COVID-19 olfactory-impaired patients who are still complaining many months after contamination, despite spontaneous olfactory recovery occurring in 40% [2] to 70% [4] of cases from 6 to 12 months. For now, only a few studies reported the OT efficiency in post-COVID-19 PPVOD. This study reports an olfactory recovery in post-COVID-19 PPVOD patients who performed ~3.5 months of OT. That olfactory recovery was significant as the SST MCID increased by more than 6 points [17] on average. Interestingly, we observed more than a doubled normosmic patients’ ratio after OT, going from 11 (25.6%) to 27 (62.6%).

We reported only a T significant improvement and normalization after OT, followed by non-significant I improvement and D worsening. This is the exact opposite of spontaneous post-COVID-19 olfactory recovery study results [30,31,32] who reported an I improvement followed by a D and, finally, a slight T improvement. A small or non-significant increasing of T was underlined by Niklassen [31], Bordin [30] and colleagues, respectively, after 4 and 6 months of spontaneous recovery. We previously confirmed these results [8] reporting that T was the most decreased olfaction subdimension as measured in a cohort of patients around 6 months after a post-COVID-19 PPVOD. 

As suggested by Iannuzzi et al. [33], spontaneous recovery in the first two months [34] could be dedicated to a significant T progression, which may correspond to early olfactory neurons and sustentacular regeneration occurring around 2 to 4 weeks in an inflammatory environment [35]. Afterward, T does not change as reported subjectively [36,37] and psychophysically [6,38] by many authors. Moreover, TDI scores seemed to better improve in patients that performed the training for more than 2 months, compared to patients with lower adherence. 

The T subdimension appeared to improve the most in compliant patients, supporting the previous discussion. Thus, there is no other potential explanation to date that could validate a spontaneous T increase after 6 months on average with persistent post-COVID-19 olfactory loss, other than an OT effect. We did not use a control group (i.e., without any OT) due to the major ethical concerns, especially in this post-COVID-19 care-seeking population, and thus we can wonder if spontaneous recovery could have produced the same results. 

Based on complete psychophysical evaluation, our normosmic population recovery proportions share some similarities to previously published cohorts who reported spontaneous recovery in 63% [7] to 73.5% [39] on average one year after the infection. Arnaud et al. [7] reported a spontaneous olfactory recovery TDI score of ~30 (as our post-OT mean TDI) 18 months after COVID-19 infection but was not peer-reviewed.

Specific to COVID-19, OT results were only reported with complete SST results in Le bon et al. [40] study that compared 10 weeks of OT with (*n* = 9) or without (*n* = 18) a 10 days oral corticosteroids course. In this study, there was no significant olfactory recovery in the OT alone group but report two nonhomogeneous groups and poor compliance for 31% of patients. Olfactory subdimensions (T, D and I) specific recoveries were not reported by authors [40]. 

In COVID-19 PPVOD, OT alone was reported as significantly improving olfaction recovery only in other steroids efficiency evaluations studies but never again with a complete SST evaluation [41,42]. However, it is recommended [15] to integrate T, D and I studies in olfactory evaluation. Indeed, OT effect on T, D and I in case of PPVOD is still unclear. Hummel firstly described a clear T increasing effect [17] of OT. So, according to our results, Oleszkiewicz et al. [26] reported a significant increasing effect on T and I in OT efficiency on post-infectious (*n* = 57) and idiopathic (*n* = 51) olfactory long-lasting dysfunctions. 

T-recovery could be explained by a peripheral regenerative [43] effect of OT with a regrowth of olfactory neurons, increase in olfactory receptor expression or a higher specific affinity for those existing as Hummel et al. [44] explained observing an improvement of electro-olfactogram after OT; and I-recovery (with D-recovery) by a more central processing allowing an olfaction dedicated area connectivity reorganization [24,45] and increase in olfactory bulbs [46]. More recently, Sorokowska et al. [25] reported, in a meta-analysis (n > 879), a large and significant post-OT increase both on D and I. However, they [25] also mentioned a small to moderate effect on T, which is in contradiction with our results despite that we used the same PEA threshold in both visits. According to these studies [17,25,26], in a PPVOD situation, we report an expected significant T-recovery compared to an insufficient I and D-recovery, with the last two typically being correlated to higher olfactory functions.

As potential neurological outcomes of COVID-19 PPVOD are becoming increasingly discussed in the literature [47,48,49], it could be an explanation for this lack of significant I-normalization and D-improvement. Moreover, the D is the only subdimension that did not significantly improve, while it is well documented that the olfactory-hippocampal network is actively involved during discrimination learning and OT [50]. Discrimination and identification tasks are closely related to cognition and specifically to executive functions, semantic task and episodic memory [51]. 

These cognitive functions might be affected by hypometabolism and dysfunction of many parts of secondary olfactory cortex areas or areas connected to them as reported in an 18FDG PET study [52] on COVID-19 PPVOD, such as the bilateral orbito-frontal cortex, cingulate gyrus, thalamus, hippocampic or parahippocampic gyri. Theses cognitive and semantic isolated impaired areas could be part of a more global connectivity structure impairment suggested in a tractography study, which is the inferior longitudinal fasciculus [53]. 

Moreover, magnetic resonance imaging morphological (and functional) modifications of many of these cortical areas, especially the gray matter volume of the cingulate gyrus and hippocampus [54], are reported to be correlated to COVID-19 persistent (≥3 months) smell loss and could be the mark of long-lasting damage to dedicated olfactory areas. Currently, there is still some doubt regarding the etiology of the central abnormalities observed as they are not proven to be the cause or the consequences of persistent olfactory loss; however, the link between D and I impairment and impaired olfactory brain areas is becoming increasingly obvious. The central involvement in post-COVID-19 persistent olfactory loss may suggest a therapeutic approach consisting of the use of cognition and semantic training that could be mediated by a speech therapist. This should be evaluated in the future.

In our study, 6 months after first evaluation, parosmia were multiplied by 3. Parosmia physiology is complex and poorly understood. It appears to be an olfactory epithelium regeneration side effect spontaneously emerging in 18% [8] to 43% [3,7] of COVID-19 PPVOD patients. Peripheral origin is supported by an abnormal neuronal regrowth, including bad proximity neuron contacts in a hypotrophic olfactory bulb environment [55]. Parosmia annoyance is not systematically correlated with olfactory function as it sometimes occurs after a total olfactory recovery in 2% [here] to 20% of cases [3]. Parosmia central origin is supported by gray matter alterations [56] and olfactory cortex hypometabolism [57]. 

Thus, the increasing parosmic patient’s ratio could be linked to peripheral regeneration induced by OT, as suggested by the significant T increase. In contrast, the persistence of this symptom could be correlated to a lack of central processing suggested by the lack of I recovery. Widely, the fact that D and I did not normalize could support a central involvement explanation for persistent post-COVID-19 olfactory loss.

Moreover, we found that parosmic patients did not lose weight, unlike non-parosmic patients. Olfaction disorder is a well-known state correlated with abnormal human control of food intake portion size, decreased reward system signals and thus satiety [58]. This study supports this effect as olfactory recovery seems to be associated with a slight but not significant loss of weight. However, in the case of weight gain during the anosmic period, when parosmia occurred, this could prevent weight loss and increase a potential metabolic risk factor associated with salt and sugar intake, which increased in nearly 30% of COVID-19 patients, especially in cases of anosmia [59].

Smell loss causes a well-known significant QoL worsening [15,60]. The benefit of such a OT in COVID 19 PPVOD is still unknown; however, we previously reported an alteration of Short-QOD-NS [8]. In this study, we reported that olfactory recovery, potentially facilitated by OT, not only induced significant improvement of Short-QOD-NS but also general QoL through SF-36 results. All Short-QOD-NS sub-scores were significantly improved but mainly one score related to social relationships. After a long olfactory deprivation, patients become used to it and develop strategies to cope with parosmia, such as avoiding tasting food or not smelling smoke. 

These behaviors generate anxiety, and patients suffer from social network reduction. Moreover, loneliness contributes to the 30% increase in depression and suicide in this specific population [61]. The emotional role and vitality SF-36 sub-domain improvements (Table 2) are consistent with the fact that olfaction is more than simply a food sense and is also a channel for social, sexual and emotional communication. Healing from an olfactory loss appears to improve the general mental state of patients. Smeets et al. [60] previously reported that all SF-36 subdimensions that improved in our study, except for general mental health, were significantly impaired in cases of severe dysosmia underlining the specific effect of OT on dysosmia-related general QoL.

We did not report any psychophysical taste evaluation even if, after 6 months, almost all patients no longer complained about it. Moreover, the small sample size reduced the study strength. Therefore, our SST subdimension OT result singularity must be confirmed with larger cohorts of patients.

## 5. Conclusions

Delayed olfaction recovery after a post-COVID-19 OT was here characterized by an increase in the psychophysical complete score (TDI) but only and significantly due to olfaction threshold normalization. These results are unusual compared to previously published post-COVID-19 olfaction spontaneous recovery studies where mainly I and D were improved. Futures studies are required on identification improvement strategies as impairment reflects a central olfactory signal misprocessing underlined here by increased parosmia.

## Figures and Tables

**Figure 1 jcm-11-03275-f001:**
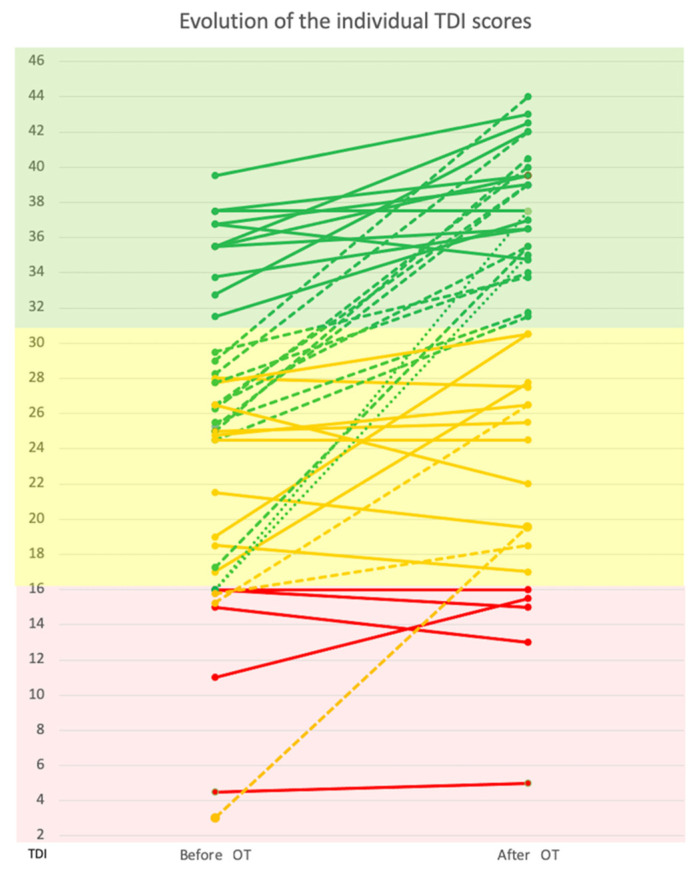
The evolution of individual TDI scores (from the SST) before and after olfactory training (OT). Colored parts cover anosmic (red), hyposmic (yellow) or normosmic (green) patients according to TDI normative values [20]. Oblique lines represent a patient anosmic (red), hyposmic (yellow) or normosmic (green) subject evolution according to post OT olfactory evaluation. Solid, dashed or pointed lines represent, respectively, patients who did not change category, changed to the upper category or changed from the anosmic to normosmic category.

**Figure 2 jcm-11-03275-f002:**
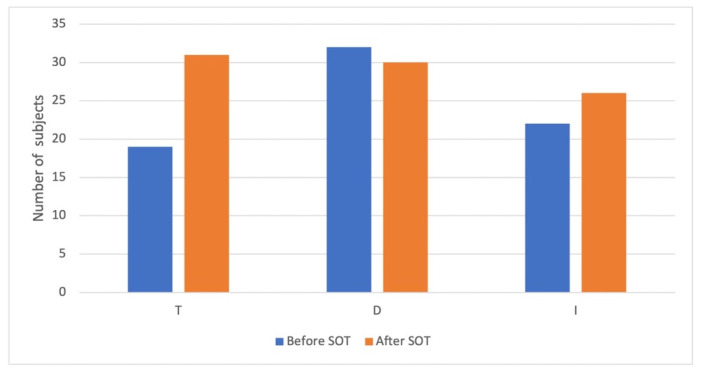
T, D and I before and after the olfactory training based on age- and sex-normalized values.

**Table 1 jcm-11-03275-t001:** Demographics and initial clinical patients feature. SD = standard deviation. * HTA = hypertension; ** GERD = gastroesophageal reflux disease; ^†^ CRASnNP = chronic rhinosinusitis without nasal polyps; and ^‡^ CRASwNP = chronic rhinosinusitis with nasal polyps.

	Mean	SD
Age (years)	41	13
Months post-COVID-19	5.8	3.2
	*n*	%
Total	43	100
**Sex**		
Female	26	61
Male	17	39
**Medical history**		
Smokers	8	18.6
Type II diabetes	2	4.6
HTA *	1	2.3
GERD **	3	7
Neurological diseases	3	7
Self-immune diseases	3	7
**Chronic rhinosinusitis**		
Allergic	16	37.2
CRSnNP ^†^	5	11.6
CRSwNP ^‡^	0	0
Neurologic diseases	3	7
**COVID-19 Severity**		
Mild to moderate illness	40	93
Severe illness	3	7
**Chemo sensorial complain**	37	86
Flavors impairment	35	81.4
Taste impairment	10	23.3

**Table 2 jcm-11-03275-t002:** Olfactory specific (Short-QOD-NS) and general (SF-36) quality of life comparative results before and after olfactory training (OT). SD represents the standard deviation. *p* represents the *p*-value at the Wilcoxon signed-rank test. * *p* < 0.05; ** *p* < 0.01; and *** *p* < 0.001.

	BEFORE OT		AFTER OT		
Quality of Life Score	Mean	SD	Mean	SD	*p*
**Short-QOD-NS**					
Total score	10.44	5.97	13.65	6.49	<0.001 ***
Social subdomain	4.58	2.7	5.88	2.86	0.001 **
Food subdomain	2.98	2.22	3.72	2.42	0.036 *
Anxiety subdomain	1.91	1.02	2.44	1.12	0.020 *
Annoyance subdomain	0.98	1.03	1.60	1.20	0.020 *
**SF36**					
Physical functioning	79.53	25.42	85.81	23.20	0.009 **
Social functioning	66.28	28.68	74.13	26.78	0.013 *
Physical role	66.28	40.42	71.51	38.80	0.407
Emotional role	55.81	41.61	64.34	42.04	0.049 *
General mental health	62.05	21.60	61.11	21.33	0.844
Vitality	44.54	24.07	51.86	21.63	0.023 *
Bodily pain	66.61	31.64	69.63	33.51	0.337
General health perception	63.63	26.09	69.56	23.22	0.045 *

## Data Availability

The data reported are part of an ongoing registration program. Deidentified participant data are not available for legal and ethical reasons. Anonymized data will be made available for research purposes, upon request.

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
