# Peer review of "Olfactory Training in Post-COVID-19 Persistent Olfactory Disorders: Value Normalization for Threshold but Not Identification"

_jcm, 2022, doi:10.3390/jcm11123275_

Round 1
Reviewer 1 Report
Dear Authors,
Thank You for submitting Your work. I read Your research with interest. Here are my comments:
- My main concern is the fact that, as You stated in the manuscript, the study lacks a control group so it is not possible to conclude that olfactory training has an effect on the olfactory recovery of patients. Also, the title of the article needs to be changed. Please revise the whole manuscript's text accordingly;
- Patient enrolled had a persistent olfactory disorder lasting from 3 to 15 months. This is a wide range and You cannot be sure that at least some patients would have recovered that sense of smell even without olfactory training. This must be stressed even more in the Discussion;
- Please be careful to include explanation for all acronyms used both in the text and in tables;
- Please consider using dots (".") instead of commas (",") for decimals;
- Did You ask patients to judge or rate their sense of smell before COVID-19? Even if it would only be a subjective evaluation and subject to bias, I would recommend to specify that all enrolled patients did not declare any olfactory impairment before COVID-19, for clarity;
- Please make sure that all sentences referring to existing literature contain the correct citation, especially in the Discussion session (eg. lines 280; 381)
- In the Methods section You state that patients underwent a 6-month olfactory training protocol, however the average timespan between the first and second evaluation was 3.5 months. Please specify the timepoints at which the patients underwent the second olfactory evaluation. If all patients were evaluated at the end of the OT, after 6 months, how is it possible that the average was 3,5 months?;
- Since the Threshold part of the TDI score was the one most significantly affected, maybe You could consider presenting a Figure just like Figure 1 but only for T scores;
- In the discussion, You state that patients gained weight during the anosmic phase, but in the Results I only read that patients lost weight between the first and second evaluation. Please comment on that and modify the Methods and Results sections accordingly;
- In the Discussion section, You state that "there are only few studies reporting spontaneous post-COVID-19 olfactory recovery". Even if only based on subjective evaluations, maybe You would find this article interesting and could add it to you bibliography: https://www.ncbi.nlm.nih.gov/pmc/articles/PMC9082048/;
- Finally, I feel that English language revision by a mother tongue speaker with expertise in scientific and medical writing is needed for improving the article, and in particular a few sentences in the Discussion session.
Thank You.
Reviewer 2 Report
The manuscript describes a prospective interventional study in post COVID-19 patients. Smell but not taste function was assessed by a validated test battery, the Sniffin in Sticks. For therapeutic intervention, olfactory training (OT) was performed in all patients and olfactory performance was re-evaluated after 6 months. The aim of the study was to determine change in olfactory function after OT. In literature there are several studies investigating OT in post COVID-19 patients but patients’ self-reports are used without using validated psychophysical testing. Therefore, the manuscript provides a major impact on the current literature. One limitation is the absence of a control arm in the study. The authors justified this lack of control with ethical issues. However, some patients started OT after 3 to 15 month (line 67). A further possibility to solve ethical issues would have been to establish a cross-over design and using a further standard therapy instead of placebo.
My comments are as follows:
The manuscript needs some minor English editing: Parosmia instead of parosmias (dysomias), olfactive impairment (line 71), There instead of These (line 205)….
Line 55: Please replace “by Pr. Hummel” with “Hummel et al” or “Hummel and colleagues” because this work was not established by a single person. The same is true for “Hummel protocol” (line 96) which was not established by a single person.
Line 55: Please add the following citation where OT was combined with low concentration odor control: Damm M, Pikart LK, Reimann H, Burkert S, Göktas Ö, Haxel B, Frey S, Charalampakis I, Beule A, Renner B, Hummel T, Hüttenbrink KB. Olfactory training is helpful in postinfectious olfactory loss: a randomized, controlled, multicenter study. Laryngoscope. 2014 Apr;124(4):826-31. doi: 10.1002/lary.24340. Epub 2013 Sep 19. PMID: 23929687.
Line 81: Please provide more information for numbers used in your formula “Compliance” (6 months, 4 weeks, and 2x7 days).
Line 108-110: Please add more information about concentration or manufactory here instead of the acknowledgement section (Payan Bertrand perfumery).
Figure 1: A TDI-score of 2 (Threshold =1 ?) is rather low in one subject. Please comment on this. Is there any compliance or mental issue in this patient?
Figure 2: Please revise the legend because there is an unnecessary repetition.
Table 2: Please explain “ET” (variance?).
Line 352: “is still unknown” instead of “in still”
Round 2
Reviewer 1 Report
Dear Authors,
Thank You for Your work on the Manuscript. I have no further comments.